# Study of the Effect of Two Phases in Li_4_SiO_4_–Li_2_SiO_3_ Ceramics on the Strength and Thermophysical Parameters

**DOI:** 10.3390/nano12203682

**Published:** 2022-10-20

**Authors:** Artem Kozlovskiy, Dmitriy I. Shlimas, Maxim V. Zdorovets, Aleksandra Moskina, Vladimir Pankratov, Anatoli I. Popov

**Affiliations:** 1Engineering Profile Laboratory, L.N. Gumilyov Eurasian National University, Nur-Sultan 010000, Kazakhstan; 2Laboratory of Solid State Physics, The Institute of Nuclear Physics of Republic of Kazakhstan, Almaty 050032, Kazakhstan; 3Institute of Solid State Physics, University of Latvia, LV-1063 Riga, Latvia

**Keywords:** blankets, lithium-containing ceramics, strength, hardness, phase composition, lithium orthosilicate

## Abstract

The paper studies the effect of Li_2_SiO_3_/Li_4_SiO_4_ phase formation in lithium-containing ceramics on the strength and thermophysical characteristics of lithium-containing ceramics, which have great prospects for use as blanket materials for tritium propagation. During the phase composition analysis of the studied ceramics using the X-ray diffraction method, it was found that an increase in the lithium component during synthesis leads to the formation of an additional orthorhombic Li_2_SiO_3_ phase, and the main phase in ceramics is the monoclinic Li_4_SiO_4_ phase. An analysis of the morphological features of the synthesized ceramics showed that an increase in the Li_2_SiO_3_ impurity phase leads to ceramic densification and the formation of impurity grains near grain boundaries and joints. During determination of the strength characteristics of the studied ceramics, a positive effect of an increase in the Li_2_SiO_3_ impurity phase and dimensional factors on the strengthening and increase in the resistance to external influences during compression of ceramics was established. During tests for resistance to long-term thermal heating, it was found that for two-phase ceramics, the decrease in strength and thermophysical characteristics after 500 h of annealing was less than 5%, which indicates a high resistance and stability of these ceramics in comparison with single-phase orthosilicate ceramics.

## 1. Introduction

Over the past few years, great attention has been paid to developments in the field of blanket ceramics, the interest in which is due to the possibilities and prospects for their use in thermonuclear energy [1,2,3]. As is known, thermonuclear energy is one of the most promising industries that can become an alternative to hydrocarbon fuel and, also, unlike nuclear energy, does not contain a large amount of nuclear long-lived waste. Thermonuclear energy is based on the mechanisms of using tritium as a nuclear fuel, which is one of the most efficient types of fuel of the near future, which underlies the fuel cycle of thermonuclear reactors [4,5].

The most promising of all currently known methods for producing tritium for its subsequent use is the method of extracting it from lithium as a result of nuclear reactions with neutrons of the Li(n, He)T type [6]. As a result of such reactions, the formation of tritium occurs, which is accompanied by a large amount of released energy, which can be transformed in the future. For the most efficie nt method of extracting and subsequent accumulation of tritium, lithium-containing ceramics are used (blankets based on compounds such as Li_2_TiO_3_, Li_2_ZrO_3_, Li_4_SiO_4_, Li_2_SiO_3_, etc. [7,8,9,10,11]). Interest in this class of ceramics is due to their high strength, mechanical, structural, and corrosion properties, as well as significant resistance to external influences, including radiation damage [12,13]. At the same time, despite a fairly large number of different studies in this direction, studies on the methods of obtaining and various modifications of lithium-containing ceramics are of great importance. Firstly, from a fundamental point of view, obtaining new data on the properties of ceramics or methods for their modification opens up great opportunities for researchers in understanding the processes of formation of materials [14,15]. Secondly, from a practical point of view, obtaining new types of ceramics with high performance and resistance to external influences opens up prospects in various technological solutions [16,17]. One of the directions in the search for optimal compositions of ceramics is the search for ways to create two-phase types of ceramics with a set of properties that significantly exceed known ceramics in terms of such indicators as resistance to external influences, thermophysical parameters, and strength characteristics [18,19,20].

The purpose of this work is to study the influence of the formation of the Li_2_SiO_3_ phase in Li_2_SiO_3_/Li_4_SiO_4_ ceramics on the strength and thermophysical parameters, as well as the resistance to long-term thermal heating. The choice of this type of ceramics is due to the possibility of creating new types of blanket materials for the production of tritium, as well as to obtaining new data on the effect of two phases on the stability of ceramics.

As a method for obtaining lithium-containing ceramics, the method of solid-phase mechanochemical synthesis was used, followed by thermal sintering of ceramics at a temperature of 1000 °C. To obtain two-phase ceramics, a variation of the initial components during mixing is used, which makes it possible to obtain a different ratio of elements which leads to the formation of impurity phases during thermal sintering.

Thus, the key hypothesis of this work is the assumption that the formation of two or more phases in the ceramic structure leads to the appearance of additional interfacial boundaries, which contributes to an increase in resistance to external influences.

## 2. Experimental Part

Chemical reagents LiClO_4_×3H_2_O and SiO_2_ of chemical purity 99.95%, manufactured by Sigma Aldrich (St. Louis, MI, USA) were used for the synthesis. For mechanochemical synthesis, a planetary mill Pulverisette 6 classic line (Fritsch, Berlin, Germany) was used. The resulting mixtures were annealed after grinding in a SNOL muffle furnace (Snol-Term, St.-Petersburg, Russia).

Table 1 shows data on the composition of the initial components used for synthesis, as well as the conditions for obtaining.

When describing the results obtained, the short sample numbers S-1, S-2, and S-3 will be used for ease of interpretation and comparison.

X-ray phase analysis of the obtained samples to determine the phase composition and parameters of the crystal structure was performed using a D8 Advance ECO X-ray diffractometer (Bruker, Berlin, Germany). The diffraction patterns were taken in the Bragg-Brentano geometry, in the angular range 2θ = 20–70°. To analyze the phase composition, the DiffracEVA v.4.2 software was used; the phases were refined by comparing the obtained diffraction patterns with the PDF-2(2016) database.

The morphological features of the synthesized ceramics were studied using scanning electron microscopy using a Hitachi TM3030 microscope (Hitachi, Tokyo, Japan).

The hardness was measured by indentation using a LECO 700 M microhardness tester with an indenter load of 100 N. The hardening of ceramics depending on the phase composition was estimated by calculating the difference between the hardness indices of the samples.

Crack resistance was determined using the single compression method at a compression rate of 0.1 mm/min. Crack resistance was determined by the maximum pressure that the ceramic can withstand under compression.

The study of the thermophysical parameters of ceramics, as well as their changes depending on the phase composition, was carried out using the method of measuring the longitudinal heat flux using the KIT-800 device (Teplofone, Moscow, Russia).

## 3. Results and Discussion

Figure 1 shows the results of X-ray diffraction of the studied ceramics, obtained depending on the variation of the components. The general view of X-ray diffraction patterns presented depending on the concentration of the components of the initial mixture indicates the processes of phase transformations with an increase in the concentration of the lithium-containing component. These phase changes are expressed as the appearance of new reflections, the presence of which indicates the formation of new phases in the ceramic structure. Below the diffraction patterns are bar graphs for the established Li_4_SiO_4_, Li_2_SiO_3_ phases and, as a comparison, a bar graph for the Cristobalite (SiO_2_) phase is shown. As can be seen from the presented bar graphs, the most probable phase for sample S-1 is the Li_4_SiO_4_ phase; however, the ratio of intensities is somewhat different than for the samples from the literature data. According to the literature data, the most intense diffraction reflections for the Li_4_SiO_4_ phase are reflections at 2θ = 27.5° and 33.5° [21,22]. In the observed diffraction pattern, the most intense reflection corresponds to the angular position 2θ = 23.3–23.5°, which may be associated with texturing processes and were also observed in [21] when obtaining lithium-containing ceramics using the combustion method. At the same time, a comparison of the position of the reflections with the Cristobalite phase showed that the position of the reflections for the experimentally obtained samples is quite different from the position of the reflections for the Cristobalite phase. Also, part of the reflections observed in the range of angles 2θ ≥ 50° is not described by the Cristobalite phase, but is in good agreement with the position of the reflections for the Li_4_SiO_4_ phase. The Figure 1 also shows X-ray diffraction patterns of the initial components LiClO_4_×3H_2_O and SiO_2_ used for the synthesis of ceramics in order to identify impurities in the synthesized ceramics. As can be seen from the presented data, the position of the main reflections of the initial components does not coincide with the position of the main observed diffraction reflections for the synthesized samples, which indicates the absence of impurity inclusions in the form of the initial components or their products in the synthesized samples. Such a strong difference in the reflection intensities for the synthesized samples can be explained by the effects associated with the texturing of the samples during synthesis and subsequent thermal sintering, which are associated with the formation of the Li_4_SiO_4_ structure by mechanochemical synthesis which can be accompanied by processes leading to the selection of a preferred direction textures in samples. Also, the manifestation of this texture can be associated with the possible formation of a substitutional solid solution or the incorporation of lithium ions into SiO_2_, followed by the formation of the Li_4_SiO_4_ phase.

For sample S-1, according to the data of X-ray phase analysis, all observed diffraction reflections are characteristic of the monoclinic phase Li_4_SiO_4_ (P21/m(11)) with crystal lattice parameters a = 5.1843 ± 0.0022 Å, b = 6.1358 ± 0.0024 Å, c = 5.3356 ± 0.0021 Å, β = 90.63° (clarification of the crystal lattice parameters was carried out using reference values from the database for the monoclinic phase PDF-01-076-1085). At the same time, diffraction reflections have an asymmetric shape, which is typical for structures containing deformation inclusions or dimensional distortions.

For sample S-2, in addition to the main reflections characteristic of the monoclinic phase, in the region of 2θ = 27–30° and 36–40°, the formation of low-intensity reflections is observed, the position of which, taking into account the refinement of the parameters, is characteristic of the orthorhombic Li_2_SiO_3_ (*Ccm21(36)*) phase, the content of which is no more than 1.5–2%. At the same time, for the main monoclinic Li_4_SiO_4_ phase, an insignificant change in the parameters of the crystal lattice is observed, which indicates the processes associated with the rearrangement of the crystal structure. The crystal lattice parameters for the Li_4_SiO_4_ phase are a = 5.1772 ± 0.0025 Å, b = 6.1250 ± 0.0031 Å, c = 5.3261 ± 0.0023 Å, β = 90.93°.

In the case of sample S-3, an increase in the intensity of reflections characteristic of the Li_2_SiO_3_ orthorhombic phase is observed, which indicates its formation with an increase in the lithium concentration in the composition of the initial mixture. An assessment of the contributions of reflections characteristic of the orthorhombic Li_2_SiO_3_ phase showed that its content increases to 20%. With such a content of the Li_2_SiO_3_ phase, the composition of the ceramics is a solid two-phase solution containing a large number of interphase boundaries, and is also characterized by an increase in the dislocation density associated with a decrease in the size of grains (crystallites, according to the assessment of structural characteristics). For the main phase of Li_4_SiO_4_, the crystal lattice parameters are a = 5.1631 ± 0.0023 Å, b = 6.1186 ± 0.0023 Å, c = 5.3179 ± 0.0021 Å, β = 90.91°.

Such phase transformations with the formation of a two-phase Li_2_SiO_3_/Li_4_SiO_4_ solution are due to a change in the lithium-containing component concentration, an increase in which leads to the initialization of phase transformation processes associated with the formation of a small amount of impurity inclusions at first, and at high lithium concentrations, the formation of a solid solution of two phases.

An analysis of the shape of the diffraction lines for samples containing the Li_2_SiO_3_ phase showed that the reflections for the main monoclinic Li_4_SiO_4_ phase become broader, while the degree of symmetry of the reflections increases, indicating structural ordering and a decrease in deformation distortions in the structure of the main phase.

According to the assessment of the width and shape of diffraction reflections using the Williamson-Hall method, it was found that the average crystallite size for sample S-1 was 33–35 nm, while for samples S-2 and S-3, the sizes of crystallites (D) were 29–30 nm and 25–26 nm, respectively. At the same time, analyzing the *βcosθ/sinθ* relation depending on the sample type, a decrease in the approximating curve slope for samples S-2 and S-3 is found, which indicates a decrease in deformation distortions of the crystal structure of these samples. Also, a change in the crystal lattice parameters of the main Li_4_SiO_4_ phase indicates a decrease in deformation and an increase in the structural ordering degree. In turn, a decrease in the grain size is accompanied by an increase in the dislocation density, as well as boundary effects associated with a decrease in grains. Using the well known expression for determining the dislocation density (*L*), which can be written as *L = 1/D^2^*, it is found that a decrease in the grain size leads to an increase in the dislocation density from 0.8 × 10^10^ cm^−2^ to 1.1 × 10^10^ cm^−2^ and 1.4 × 10^10^ cm^−2^ for samples S-1, S-2, and S-3, respectively. Such an increase in the dislocation density, combined with a decrease in deformation distortions can lead to the strengthening of ceramics under external influences and an increase in resistance to mechanical damage.

Figure 2 shows the SEM images of the studied ceramics, reflecting the morphological features of the synthesized samples depending on their phase composition.

As can be seen from the presented SEM images, in the case of sample S-1, the grain structure has an approximate diamond shape, while the grain boundaries are quite large. Also, the grain morphology for sample S-1 is quite complex and has a large number of faces, as well as grains of irregular shape. For sample S-2, the presence of grains of two types is observed, which have significant differences both in morphological features and size. Figure 2b clearly shows the presence of smaller grains (no larger than 10 nm) located near the joints and grain boundaries, the presence of which may be due to the formation of the Li_2_SiO_3_ phase. At the same time, for sample S-3, which, according to the data of X-ray phase analysis, an increase in the contribution of the Li_2_SiO_3_ phase is observed; the number of impurity grains near the boundaries of large grains increases significantly, which confirms the above assumption that these grains correspond to the Li_2_SiO_3_ phase. An analysis of morphological features depending on the Li_2_SiO_3_ phase contribution concentration showed that an increase in the content of the Li_2_SiO_3_ phase leads to a denser packing of ceramics, as well as an increase in grain boundaries in which impurity particles are located. At the same time, for sample S-3, there are practically no large separations between grains that are present on the surface of sample S-1 and, to a lesser extent, on sample S-2.

To determine the elemental composition and the presence of impurity phases, the method of energy dispersive analysis by mapping is applied. In view of the lack of the possibility of detecting lithium by the method of energy dispersive analysis, the determination of the phase composition is carried out by a comparative analysis of the Si:O ratio, which has a different ratio for each of the established phases. Figure 2 also shows the results of mapping, which reflect the distribution of impurity phases in the structure. The general trend of the observed changes in impurity inclusions reflects an increase in their concentration and a more complex structure.

As is known, the choice of blanket materials for tritium propagation is based on the combination of properties of lithium-containing ceramics, the variation of which makes it possible to obtain the most efficient breeders. So, for example, the choice of lithium-containing ceramics based on Li_4_SiO_4_ is primarily due to the density of lithium, as well as the resistance to mechanical stress. At the same time, the thermomechanical stability is the best for ceramics based on Li_2_TiO_3_ and Li_2_ZrO_3_. At the same time, during analysis of the thermophysical and mechanical properties of ceramics, as well as resistance to external influences, much attention is paid to grain sizes, material density, and phase composition, and the study of the phase composition, as well as the possibility of obtaining two-phase ceramics and their properties, is the most promising area of research in the field of blanket materials. For example, it was shown in [23] that the formation of two-phase ceramics leads to an increase in resistance to crack resistance under single compression, which is associated not only with the dual composition of ceramics, but also with grain size effects. The influence of the size effect on the resistance to external mechanical influences, as well as to the compressive load, was reported in [24]; according to the presented data, a change in the grain size leads to the strengthening of ceramics, as well as an increase in the crack resistance. It was also shown in [25] that, in the case of Li_4_SiO_4_/Li_2_SiO_3_ two-phase ceramics, coarsening of the grain sizes and, consequently, an increase in porosity, leads to a sharp decrease in crack resistance by a factor of 1.5–2. Also, the presence of the second Li_2_O phase in the structure of Li_4_SiO_4_ ceramics, according to the data of [26], leads to the strengthening of ceramics up to 60–80% compared to single-phase ceramics.

Figure 3 shows the results of changes in the hardness and resistance to single compression of ceramics depending on the phase composition of the samples. The stability results are compared with a number of literature data taken from [25,26,27].

The general view of the trend in the change in strength characteristics indicates a positive effect of the formation of the Li_2_SiO_3_ impurity phase on the composition of ceramics. As can be seen from the presented data, the formation of the Li_2_SiO_3_ phase with a content of more than 3% leads to an increase in hardness and crack resistance by more than 20–30% in comparison with single-phase Li_4_SiO_4_ ceramics. At the same time, the presence of the Li_2_SiO_3_ impurity phase leads to a more effective increase in crack resistance during single compression than during hardness indentation. An increase in the Li_2_SiO_3_ phase contribution to 20% leads to a strengthening of ceramics by more than 65% and an increase in crack resistance by more than two times (data for sample S-3). At the same time, a comparative analysis of crack resistance with literature data showed that two-phase ceramics significantly exceed the performance of similar samples of orthosilicate ceramics, and the established effect of the Li_2_SiO_3_ impurity phase on the increase in strength characteristics is in good agreement with the literature data.

The effect of strengthening ceramics can also be explained by the inclusion of the Li_2_SiO_3_ phase in the composition of ceramics, and also by a change in the dislocation density, as well as size and grain boundary effects. Evaluation of the effect of grain boundaries and their volumetric contribution (*f_gb_*) was evaluated using Formula (1):(1)$$f_{gb}=\frac{3\delta \left(D^{2}-\delta^{2}\right)}{D^{3}},$$
where *D* is the grain size and *δ* is the grain boundary size. The results of determining this value are shown in Figure 4. Evaluation results of the volumetric contribution of grain boundaries and dislocation density depending on the change in grain size to hardening and increase in the crack resistance of ceramics are shown in Figure 4. As can be seen from the presented SEM images (see data in Figure 2), a change in the phase composition leads to a decrease in the grain size, and as a result, an increase in the proportion of grain boundaries, near which impurity inclusions are formed in the form of fine grains. Thus, a change in grains and grain boundaries can lead to an increase in the resistance to the propagation of microcracks under external action, and an increase in dislocation density leads to a slowdown in the processes of propagation in depth along grain boundaries. In turn, the data presented in Figure 4 are in good agreement with the above assumption about the positive effect of the influence of the size factor and changes in the dislocation density on the strengthening of ceramics.

An important characteristic of the use of lithium-containing ceramics as blanket materials, in addition to mechanical properties, is their thermophysical parameters, which determine the resistance of ceramics to thermal heating, as well as the ability to transfer heat.

Figure 5 shows the results of changes in the thermal conductivity and thermal diffusivity for the samples under investigation, measured in the temperature range of 300–1000 K, reflecting changes in the heat-conducting properties of ceramics. The general view of the presented data indicates that a change in the phase composition of ceramics leads to an increase in thermal conductivity, as well as the preservation of its stability in a wide temperature range. In turn, for samples S-2 and S-3, changes in the thermal conductivity in the entire measured temperature regime are practically not observed, with only a small exception in the temperature range of 300–500 K, for which there is an insignificant decrease in thermal conductivity, which amounted to no more than 1–3%. In turn, for the thermal diffusivity value, a decrease is observed in the temperature range of 300–500 K, with subsequent stabilization of the value after 600 K, indicating stabilization and resistance of ceramics to thermal loads.

Figure 6 shows the results of changes in the thermophysical parameters of the studied ceramics depending on the phase composition, which change with variation of the initial components. The general view of changes reflects the effect of the impurity phase on the increase in thermal conductivity and thermal diffusivity. Also, the obtained values were compared with the literature data on thermophysical values for single-phase Li_2_TiO_3_ and Li_4_SiO_4_ ceramics. For comparison, the results of changes in the values of thermal conductivity and thermal diffusivity taken from [28] were chosen.

As can be seen from the data presented, the appearance of the Li_2_SiO_3_ phase leads to an increase in thermal conductivity by 18%, and an increase in its contribution to 20% leads to an increase in thermal conductivity up to 40%, and an analysis of the temperature dependence of the change in the thermal conductivity showed that the formation of an impurity phase leads to an increase in stabilization changes in thermophysical properties over the entire measured temperature range. Comparison of the obtained thermophysical parameters with the literature data showed that the proposed method for obtaining ceramics, as well as their phase composition variation, is quite effective, which is expressed in a significant increase in thermophysical properties in comparison with the ceramics obtained in the work [28].

Figure 7 shows the results of testing the synthesized ceramics for resistance to long-term thermal heating at a temperature of 700 °C for 500 h. After every 50 h, the samples were tested for crack resistance and determination of the thermal conductivity coefficient.

The general trend of change in resistance to cracking as a result of prolonged thermal heating indicates that the presence of Li_2_SiO_3_ impurity phases leads to an increase in resistance to high-temperature degradation, which is most pronounced for single-phase ceramics. The most pronounced changes in crack resistance occur after 250 h of testing, which indicates that the degradation processes are cumulative, and the nature of the change after 250 h is non-linear for samples of single-phase ceramics (S-1), in contrast to sample S-3. At the same time, the results for sample S-3 after 500 h of testing showed that the decrease in crack resistance was less than 5%, which is within the limits of acceptable changes.

The results of the thermal conductivity change shown in Figure 7b indicate the effect of thermal degradation on the thermal properties of ceramics. At the same time, as in the case of changes in the indicators of strength characteristics, a positive effect of the presence of impurity phases in the composition of ceramics on an increase in the resistance to degradation of the thermophysical properties of ceramics was established.

## 4. Conclusions

The work is devoted to the study of the properties of lithium-containing ceramics based on orthosilicates obtained by mechanochemical synthesis followed by thermal annealing, as well as the variation of the initial components during synthesis. During the studies, it was found that the variation of the ceramic components leads to a change in the phase composition, as well as the morphological features of the synthesized samples. According to the data of X-ray phase analysis, it was found that an increase in the lithium concentration in the composition of the initial mixtures leads to the formation of an impurity Li_2_SiO_3_ phase, followed by the formation of a solid two-phase solution of Li_2_SiO_3_/Li_4_SiO_4_. During determination of the strength and thermophysical parameters, a positive dynamic of hardening and an increase in the heat-conducting properties of ceramics during the formation of impurity phases in them was established.

Further research in this direction will be aimed at the study of the radiation damage resistance of ceramics and assessment of the kinetics of radiation defect accumulation in the near-surface layer of ceramics.

## Figures and Tables

**Figure 1 nanomaterials-12-03682-f001:**
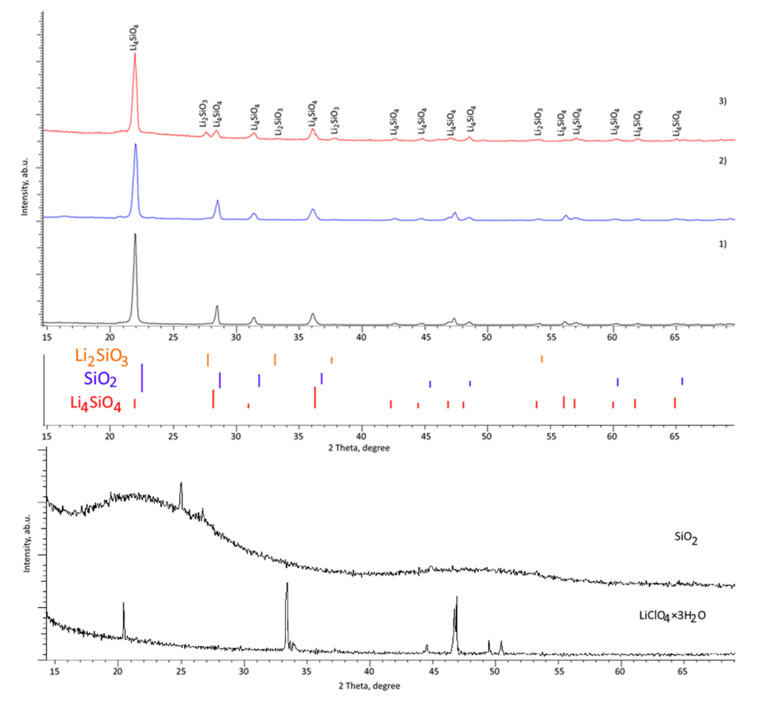
Results of X-ray diffraction of the studied samples obtained at different concentrations of the LiClO_4_:SiO_2_ components: (1) 0.1:0.9 (2) 0.25:0.75; (3) 0.5:0.5 (the figure shows the bar graphs of the positions of the diffraction reflections of the reference values for various phases, as a comparison, the positions of the main lines for the Cristobalite (SiO_2_) phase are shown). X-ray diffraction patterns of the initial components LiClO_4_×3H_2_O and SiO_2_ are presented.

**Figure 2 nanomaterials-12-03682-f002:**
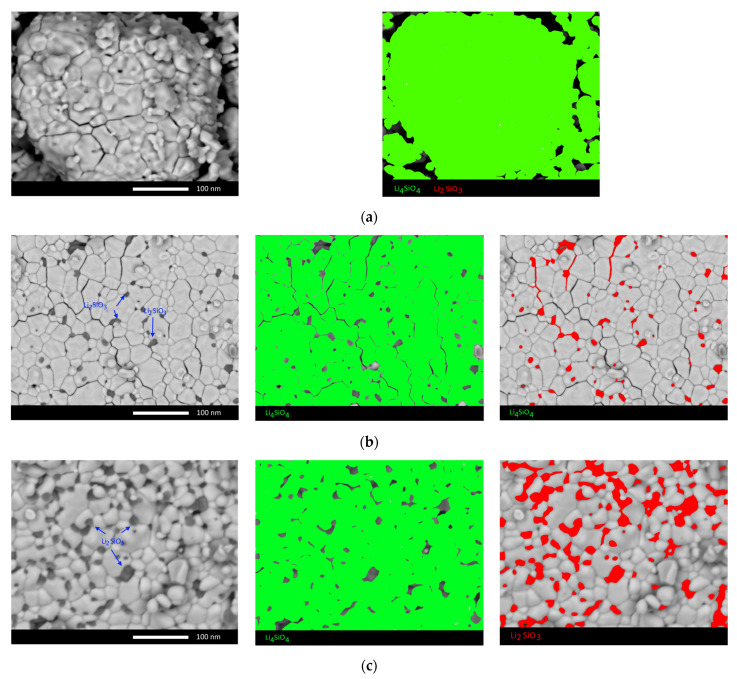
SEM images of the studied ceramics: (**a**) S-1; (**b**) S-2; (**c**) S-3.

**Figure 3 nanomaterials-12-03682-f003:**
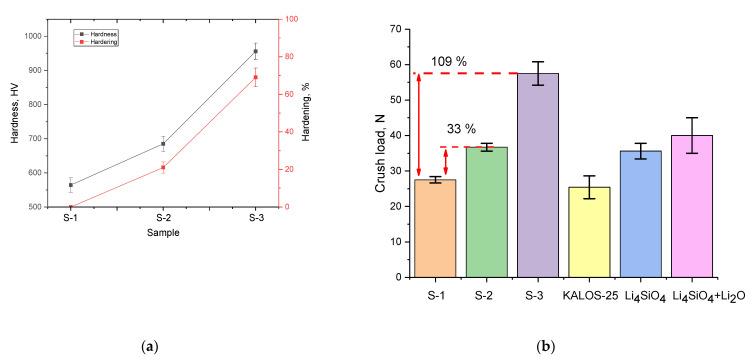
(**a**) Results of changes in the hardness of ceramics during indentation; (**b**) results of changes in the single compression resistance value.

**Figure 4 nanomaterials-12-03682-f004:**
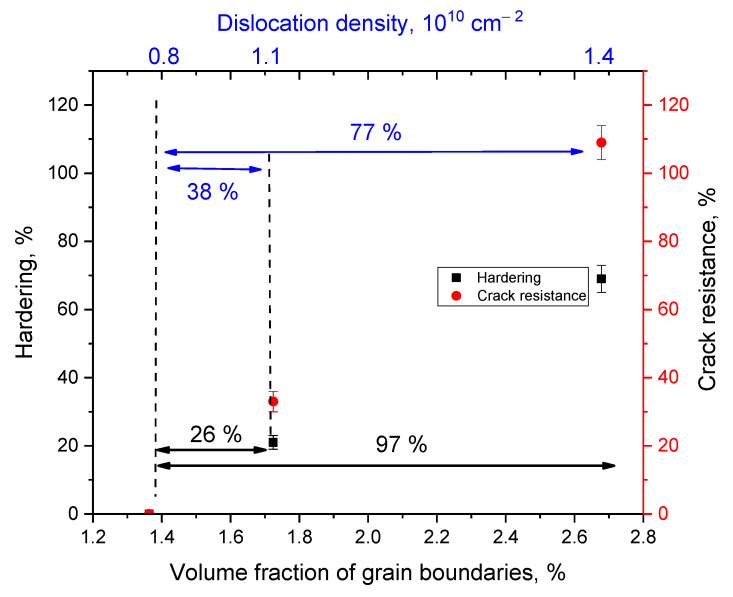
Evaluation of the contributions of changes in the dislocation density and volume fraction of grain boundaries in ceramics to the effects of hardening and increase in crack resistance.

**Figure 5 nanomaterials-12-03682-f005:**
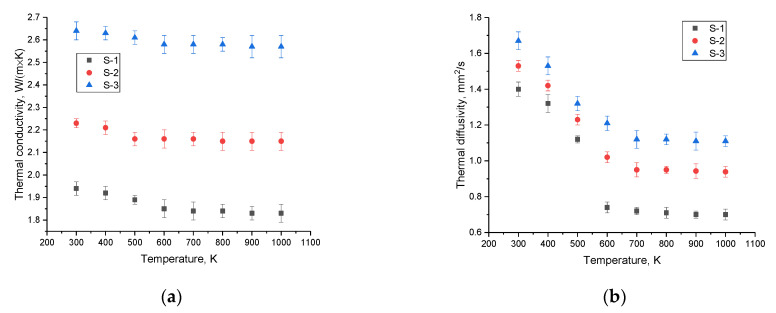
(**a**) Results of changes in thermal conductivity for the studied samples of ceramics; (**b**) results of changes in thermal diffusivity for the studied samples of ceramics.

**Figure 6 nanomaterials-12-03682-f006:**
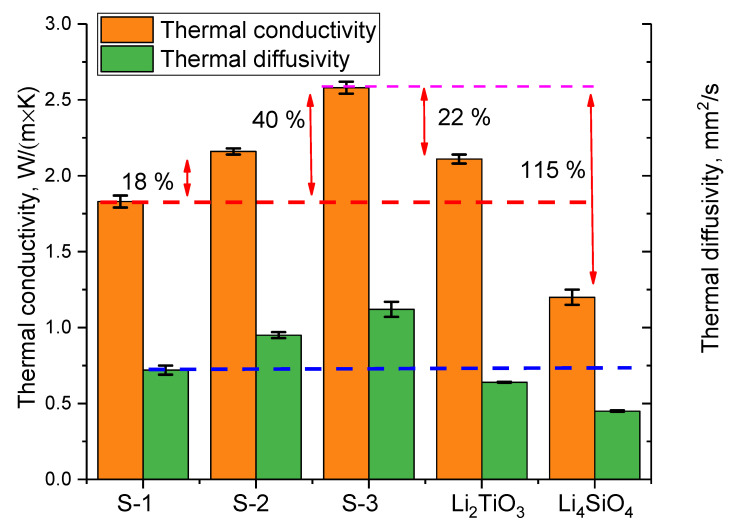
Results for the comparison of thermophysical parameters on the studied ceramics.

**Figure 7 nanomaterials-12-03682-f007:**
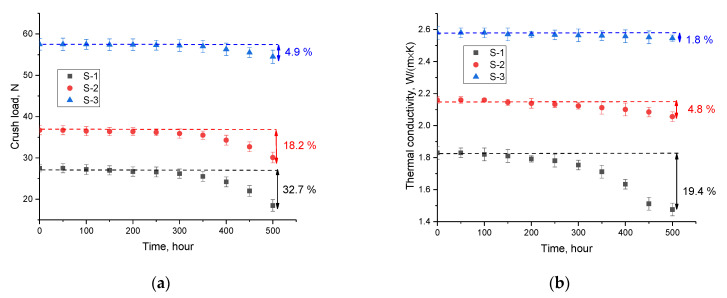
(**a**) Results of the stability of crack resistance during long-term thermal heating; (**b**) the results of the change in the thermal conductivity coefficient depending on the time of thermal tests.

**Table 1 nanomaterials-12-03682-t001:** Data of experiments on obtaining lithium-containing ceramics.

Sample	Component Content, mol		Synthesis Conditions
	LiClO_4_×3H_2_O	SiO_2_	
S-1	0.10	0.90	1. Mechanical grinding in a planetary mill at 250 rpm for 30 min. 2. Thermal sintering in a muffle furnace at a temperature of 1000 °C for 8 h, followed by cooling for 24 h
S-2	0.25	0.75	
S-3	0.50	0.50	

## Data Availability

Not applicable.

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
