# Peer review of "Study of the Effect of Two Phases in Li4SiO4–Li2SiO3 Ceramics on the Strength and Thermophysical Parameters"

_nanomaterials, 2022, doi:10.3390/nano12203682_

Round 1

Reviewer 1 Report

The authors studied the influence of the formation of the Li2SiO3 59 phase in Li2SiO3/Li4SiO4 ceramics obtained by mechanochemical synthesis followed by thermal annealing on the strength and thermophysical parameters as well as the resistance of the ceramics to long-term thermal heating. Using the X-ray phase analysis, they have found that an increase in the lithium concentration in the composition of the initial mixtures leads to the formation of an impurity Li2SiO3 phase, followed by the formation of a solid two-phase solution of Li2SiO3/Li4SiO4. Positive dynamics of hardening and an increase in the heat-conducting properties of the ceramics during the formation of impurity phases in them have been revealed.
The manuscript has a purely technological character, so a deeper physical analysis would strongly contribute to its significance. However, it is only a suggestion.
There are, however, a few faults that must be rectified before publication:
- Eq. (1) is introduced but after that, it is not used in the following discussion.
- The legend in Fig. 7b is missing.
After considering the above recommendations, the manuscript may be published in Nanomaterials.

Author Response

Dear Reviewer,

Please see our reply in attached file:

Reviewer 2 Report

I think the phase identification is incorrect and the main phase is not Li4SiO4 but cristobalite. The XRD pattern presented in Fig. 1 does not match that of Li4SiO4 presented in many papers devoted to synthesis of Li4SiO4 but it is just the XRD pattern of cristobalite. The diffraction patterns were taken in the angular range 2θ=20-70⁰. I recommend to record the XRD patterns starting from 2θ=10⁰. It will help to confirm the correctness of the phase identification.

I think it is not reasonable to discuss the other results of the study until the phase identification is clarified.

Only one remark on the list of references. Some important references are missed. The list of references should be updated. The list of references should be prepared in the same style.

Author Response

(The authors gave the same response as above.)

Round 2

Author Response

Editor Decision:

The Reviewer2 concerns have been addressed and the manuscript was improved accordingly. However the ar still missing points in strutcural investigation. E.g in Fig. 2 SEM investigations EDS of elements would be requiered not only point to some black points to show a phase.

Authors response:

The authors express their gratitude to the reviewer for this decision and for taking into account all the corrections in the first round of peer review. In response to comments regarding EMF, the results are shown in Figure 2 as phase mapping data.